# The impact of multi-decadal changes in VOCs speciation on urban ozone chemistry: A case study in Birmingham, United Kingdom.

Jianghao Li[1,2], Alastair C. Lewis[1,3], Jim R. Hopkins[1,3], Stephen J. Andrews[1,3], Tim Murrells[4], Neil Passant[4], Ben Richmond[4], Siqi Hou[5], William J. Bloss[5], Roy M. Harrison[5,6], Zongbo Shi[5].

[1]Wolfson Atmospheric Chemistry Laboratories, University of York, York YO10 5DD, UK

[2]School of Water and Environment, Chang'an University, Xi'an 710064, China

[3]National Centre for Atmospheric Science, University of York, Heslington, York YO10 5DD, UK

[4]Ricardo Energy and Environment Gemini Building, Fermi Avenue, Harwell, Oxon OX11 0QR, UK

[5]School of Geography, Earth and Environmental Sciences, University of Birmingham, Edgbaston, Birmingham B15 2TT, UK

[6]Department of Environmental Sciences, Faculty of Meteorology, Environment and Arid Land Agriculture, King Abdulaziz University, P.O. Box 80208, Jeddah 21589, Saudi Arabia

*Correspondence to:* Jianghao Li (cfm531@york.ac.uk)

**Abstract** Anthropogenic non-methane volatile organic compounds (VOCs) in the United Kingdom have been substantially reduced since 1990, in part attributed to controls on evaporative and vehicle tailpipe emissions. Over time other sources with a different speciation, for example alcohols from solvent use and industry processes, have grown in both relative importance and in some cases in absolute terms. The impact of this change in speciation and the resulting photochemical reactivities of VOCs are evaluated using a photochemical box model constrained by observational data during a summertime ozone event (Birmingham, UK), and apportionment of sources based on the UK National Atmospheric Emission Inventory (NAEI) data over the period 1990-2019. Despite road transport sources representing only 3.3% of UK VOC emissions in 2019, it continued as the sector with the largest influence on local $O_3$ production rate ($P(O_3)$). Under case study conditions, the 96% reduction in road transport VOC emissions that has been achieved between 1990 – 2019 has likely reduced daytime $P(O_3)$ by ~1.67 ppbv h$^{-1}$. Further abatement of fuel fugitive emissions was modeled to have had less impact on $P(O_3)$ reduction than abatement of VOCs from industrial processes and solvent use. The long-term trend of increased emissions of ethanol and methanol have somewhat weakened the benefits of reducing road transport emissions, increasing $P(O_3)$ by ~0.19 ppbv h$^{-1}$ in

the case study. Abatement of VOC emissions from multiple sources has been a notable technical

and policy success in the UK but some future benefits (from an ozone perspective) of the phase out

of internal combustion engine passenger cars may be offset if domestic and commercial solvent use

of VOCs were to continue to increase.

## 1. Introduction

Elevated tropospheric ozone ($O_3$) has been a long-standing pollutant of concern in the rural

and sub-urban environment and is now becoming more prevalent in urban centers as primary NO

traffic emissions reduce (Sicard, 2021). As an important tropospheric oxidant and greenhouse gas

(Kumar et al., 2021), exposure to $O_3$ also increases risks of mortality from respiratory diseases and

adversely impacts on crop productivity (Lefohn et al., 2018). $O_3$ is mainly formed through

photochemical reactions involving the oxidation of volatile organic compounds (VOCs) in the

presence of nitrogen oxides ($NO_x$, $NO_x=NO+NO_2$) (Calvert et al., 2015). The release of VOCs arises

from a wide range of activities, including unburned fuel or partially combusted products in exhaust,

from solvents used in industry and numerous other diffuse domestic and commercial sources (He et

al., 2019). Effective policies to mitigate ozone pollution rely on an accurate estimate of both

emissions and speciation of $O_3$ precursors.

The challenge in reducing $O_3$ lies in its non-linear relationship with its precursors since

individual VOCs have unique capacities for forming ozone. Decades of modelling studies have

established regimes where reductions in $NO_x$ or VOCs emissions would be preferentially beneficial

to mitigate $O_3$ – so-called $NO_x$-limited or VOC-limited regimes (Seinfeld and Pandis, 2016).

Abatement of VOCs sources is important in VOCs-limited areas since decreasing the emissions can

effectively reduce the local $O_3$ production rate and help limit $O_3$ peak concentrations (Gaudel et al.,

2020). The wide range of sources, including many that are diffuse and occur indoors, and differing

photochemical reactivities further complicates $O_3$ reduction strategies. Different mixes of sources

and speciation can lead to a need for localized policies. For example, short-chain alkanes and

alkenes with high hydroxyl radical reactivity emitted from on-road transportation in China, have

been reported as being responsible for 26% of national $O_3$ formation (Wu and Xie, 2017). A field

observation study in Delhi, India reported that the $O_3$ production in that city was most sensitive to

monoaromatics, followed by monoterpenes and alkenes during a post-monsoon period in 2018
(Nelson et al., 2021). Another study at urban sites in Seoul, South Korea concluded that the $O_3$
production was controlled by C>6 aromatics and isoprene during a summer $O_3$ episode in 2016
(Schroeder et al., 2020). There is therefore no 'one size fits-all' policy in terms for which sources
and sectors to target for optimal $O_3$ abatement efforts.

Policy and regulation aimed at improving air quality in many countries including the United
States, the United Kingdom and Europe have led to decades of falling VOCs emissions (Lewis et
al., 2020; Coggon et al., 2021). This reduction can be substantially attributed to the successful
technical implementation of tailpipe exhaust after-treatment technology for gasoline vehicles
controls on evaporative emissions from vehicles including during re-fueling and a more widespread
set of efforts to control industrial emissions (Winkler et al., 2018). Despite these successes, $O_3$
remains a pollutant of concern; whilst peak concentrations during $O_3$ events have reduced in the UK,
increases in the long-term urban background $O_3$ concentrations have been observed since the 1990s
(Department for Environment, Food & Rural Affairs, 2023). A variety of explanations have been
given to account for the increase including a rising northern hemisphere background $O_3$, increasing
methane which contributes to both global radiative forcing and enhances $O_3$ production (Tarasick et
al., 2019; Abernethy et al., 2021), the increases in non-vehicular sources of VOCs emissions
(Mcdonald et al., 2018; Yeoman and Lewis, 2021), and the reduction of $NO_x$ in VOCs-limited urban
areas leading to greater $O_3$ production efficiency (Sicard et al., 2020).

The UK National Atmospheric Emissions Inventory (NAEI) for VOCs has shown increases in
the relative contribution of solvent usage and the food & wine industry to total national VOCs
emission over 1990-2019, and steady growth in the relative importance of OVOCs within the overall
speciation (Lewis et al., 2020). In North America and Europe cities, OVOCs emitted from volatile
chemical products (VCP) can outweigh fossil fuel sources for urban VOCs. Modelling results
showed that the additional OVOCs from VCP emissions were the most important species for urban
$O_3$ production, increasing the daily maximum $O_3$ mixing ratio by as much as 10 ppbv in Los Angeles
(Qin et al., 2021). Substantial OVOCs emissions can come from unexpected places. For example,
alcohols emitted from use of windshield fluid are now estimated to be a larger VOC source from
road transport than VOC from the tailpipe in the UK (Cliff et al., 2023). From the perspective of $O_3$
pollution, the benefit of substantial reductions on vehicle emissions, whilst there has been a parallel

increasing role for non-industrial solvent usage remains unclear. What effect this shift in speciation
is having on ozone chemistry is less well studied. One challenge has been the lack of routine
measurement of OVOCs in most national air quality monitoring networks (Air Quality Expert
Group, 2020).
Recent model analysis of decadal trends of ozone has centred on the association between
extreme weather and ozone events and projected changes in ozone concentration under chemical
regime change scenarios. Significant decline in the UK experiencing ozone episodes and increase
in the background ozone concentration are found over the past three decades (Diaz et al., 2020).
Elevated ozone mostly occurs in the late spring and summer during anticyclonic conditions when
slow moving air masses from continental Europe contribute to accumulation of precursor emissions
and enhance photochemical production of ozone (Hertig et al., 2020; Lewis et al., 2021). Higher
temperatures in the late summer increase biogenic VOC emissions and reduce ozone deposition,
leading to summertime maximum concentrations (Finch and Palmer, 2020). Several studies pointed
out that increasingly hot summer due to climate change may offset gains made in the reduction of
ozone event over time (Gouldsbrough et al., 2022; Liu et al., 2022). In term of ozone production
sensitivity, reductions in NO emissions have led to decreasing trend in annual average concentration.
However, 20% or 40% $NO_x$ emission reductions would lead to increases in average and maximum
ozone concentrations in the UK with respect to 2018 (Gouldsbrough et al., 2024). In addition to
results on $NO_x$ and VOCs sensitivities, Ivatt et al. (2022) revealed that ozone concentrations in North
America and Europe were inhibited by aerosol in the 1970s, and this 'aerosol-inhibited regime' has
been shifted to Asia by 2014.
There have been several recent studies focussed on evaluating the impacts of reductions in
anthropogenic sources on ozone production. By integrating the U.S. fuel-based inventory of vehicle
emissions (FIVE) into air quality model, McDonald et al. (2018) assessed ozone sensitivity to
mobile source $NO_x$ emissions over the Eastern U.S., and Coggon et al. (2021) employed FIVE with
volatile chemical products (VCPs) to evaluate contributions of VOC from fossil fuel and different
types of VCP emissions to ozone production at an urban background site in New York City. Nelson
et al. (2021) used emission inventories from Emission Database for Global Atmospheric Research
(EDGAR) to investigate *in -situ* ozone production sensitivity to five inventory source sectors at an
urban site in Delhi. Kang et al. (2022) applied two emission inventories in air quality model to
evaluate contributions of industry, road transport, power plant, and biogenic emissions to ozone
production in Chinese cities. Although great efforts have been made toward identifying crucial VOC
sources in regional ozone production, understandings of their roles in urban ozone chemistry in the
context of historical changes is still lacking.
In this study we evaluate the effects of changing speciation on urban ozone chemistry, using
recent field measurements of $O_3$ and its key precursors such as $NO_x$, CO, speciated VOCs and
OVOCs in Birmingham, UK during August 2022. We combine this with changing speciation and
relative amounts of VOCs based on long-trends in the NAEI. The sensitivity of *in- situ* production
and OH reactivities of the measured $O_3$ precursors are investigated by constraining the observational
data sets to a zero-dimensional chemical box model. By incorporating the detailed NAEI VOCs
emission inventories over the period of 1990-2019 into the model, $O_3$ formation in Birmingham is
used as a case study to quantify the impacts of the real-world changes in VOCs sources on urban $O_3$
production rate. The relative importance of different VOCs functional group classes to $O_3$
production are also evaluated. The results help understand impacts of decades of abating different
VOCs-emitting sectors on urban $O_3$ production, and outline the implications for future $O_3$ control
strategies.

## 2.   Materials and Methods

2.1 Field observations
The observations are taken from the Birmingham NERC Air Quality Supersite during August
2022. This is located on the University of Birmingham (52°27'20.2"N 1°55'44.3"W) campus. The
site has been in operation for many years, and represents an urban background environment. It is
influenced by transport emissions from nearby arterial roads and residential emissions from
surrounding area. There are no significant industrial activities within a 4km radius of the site.
Continuous measurements of NO, $NO_2$, CO, $CH_4$, VOCs, $O_3$, along with meteorological
parameters including air temperature and pressure, relative humidity, wind speed and direction were
made. Briefly, NO and $NO_2$ were measured by a chemiluminescence-based T200 analyzer (Teledyne
API., U.S.A.) and the T500U Cavity Attenuated Phase Shift (CAPS) analyzer (Teledyne API.,
U.S.A.). The concentration of $NO_x$ was then the statistical sum of NO and $NO_2$. The mixing ratio of
CO were measured by a laser absorption spectroscopy Multi-species Continuous Emissions
Monitoring instrument (Enviro Technology Service Ltd., UK) (Li et al., 2020). Manual calibration
and span checks for the above instruments were performed every 3 days, and automatic zero
calibration was set on daily bases. $O_3$ was measured by an $O_3$ analyzer (Model 49i, Thermo Fisher
Scientific Inc., U.S.A.) with a minimum detection limit (MDL) of 1.0 ppbv. Meteorological
parameters including air temperature and pressure, and relative humidity were obtained from a
weather station WS300-UMB weather station (Luff GmbH, Germany). Additionally, Wind speed
and direction were measured by a 3-axis ultrasonic anemometer (Gill Instruments Ltd., UK) over
the campaign.
A gas chromatography-flame ionization detection (GC-FID) analysis system (7890A, Agilent
Technologies, U.S.A.) was used to quantify 38 individual VOCs species. Details on instrument
settings and quality assurance/quality control methods can be found in (Warburton et al., 2023).
Briefly, the GC-FID system utilizes dual detectors: one detector for $C_2 - C_6$ non-methane
hydrocarbons (NMHCs);the other detector for remaining $C \geq 7$ hydrocarbons, and polar species
such as ethers, ketones, and alcohols. Ambient samples were dried at $-40\ °C$ using a water trap and
then preconcentrated on a carbon adsorbent at the lowest temperature the unit could achieve, always
lower than -115 °C. Once a 0.5L sample had been collected, a pre-concentration trap was warmed
slightly from -80 °C to purge trapped atmospheric $CO_2$. The trap was then heated to 190 °C for 3
minutes with a counter flow of helium thermally desorbing the concentrated VOCs onto focusing
micro-trap held at lower than -115 °C. The analytes were flash heated and passed onto a VF-WAX
column. The unresolved analytes ($C_2 - C_6$ NMHCs) were then transferred into a $Na_2SO_4$-deactivated
$Al_2O_3$ porous-layer open tubular (PLOT) column via a Deans switch, for separation and detection
by the first FID. The Dean switch then diverted the analytes onto a fused silica internal diameter to
balance column flows and subsequently transfer the VF-WAX column- resolved species into the
second FID. Generally, quantification of $C_2$-$C_6$ hydrocarbons was completed by the first FID using
4 ppbv gas standard cylinders (the National Physical Laboratory, Teddington, UK). Quantification
of $C \geq 7$ hydrocarbons and OVOCs was completed by the second FID using effective carbon number
(ECN) with reference to toluene. In this study, the concentration of total VOC (TVOC) was defined
as the statistical sum of concentrations of measured individual species, but this is not meant to infer
that this represents the total reactive carbon in air, which would always be greater than this value
due to unmeasured species. Later in this study we broadly group species according to their chemical
function groups, summing into alcohols, ketones, alkanes, alkenes, aromatics, aldehydes, and
alkynes.
The GC-FID system responses were regularly checked by running direct calibration sequences
using the 4 ppbv gas standard cylinders. It was verified there was no FID-response drift over the
analyzing period for this study. Additionally, carbon-wise FID responses for all reported species
were calculated to verify the use of ECN as a quantification method. Table S1 lists which species
were directly calibrated, and which used equivalent carbon numbers for quantification. Table S2
shows effective carbon numbers of species which used carbon-wise responses.
2.2 National emission inventory for VOCs
Estimates of UK anthropogenic VOC emissions are taken from the NAEI. The NAEI uses a
combination of UK-specific methods and default methods as recommended in the European
Monitoring and Evaluation Programme (EMEP)/European Environment Agency (EEA) Emission
Inventory Guidebook (European Environment Agency, 2016). Further details can be found in (NAEI,
2021). The VOC inventory is also disaggregated into inventories for each individual VOC species
and details of the speciation process and assumptions can be found in (Passant, 2002) and (Lewis et
al., 2020).
Methods to estimate emissions can be divided into two groups: those using emission factors,
and those using 'point source' emissions data reported to regulators by the operators of individual
industrial sites. The emission factor methods require UK activity data, for example consumption of
paint, consumption of a fuel, production of steel or vehicle kilometers travelled. The activity data is
then combined with an emission factor which expresses the total VOC emission that is expected per
unit of a given activity. Most total VOC emission factors are taken from the internationally applied
EMEP/EEA Emission Inventory Guidebook and so are not necessarily UK-specific. The factors for
road transport are directly calculated for the UK and a particularly detailed approach is used to
estimate emissions using emission factors from the Guidebook for many different vehicle types and
emission standards, fuels and road types combined with detailed transport activity from the UK
Department for Transport. Government statistics cannot always provide the necessary activity data
for other sectors, so industry data are used instead. For instance, NAEI data on consumption of
products containing organic solvents are from industry sources. The alternative point source method
can be used for source categories where emissions data can be obtained for all sites within the sector,
and this limits the method to source categories such as crude oil refining, steel production and
chemicals production. The emissions data reported by the operators of these sites can be based on
emissions monitoring, although this is not always the case and emissions might instead be estimated,
for example, using emission factors.
The NAEI produces updates to the inventory for total VOC mass emissions by source sector
each year to achieve a consistent historic time-series reflecting trends in UK emissions. Emissions
of individual VOC species are estimated using source-specific speciation profiles which show the
mass fraction of each species, or in some cases groups of species, emitted by the source (NAEI,
2021; Passant, 2002). Over 600 individual VOC species or species groups are included in the
speciation, based on sources in industry, regulators and in some cases literature sources and
databases such as the USEPA SPECIATE database. The speciated inventory tends to be more
uncertain than the estimation of total mass of VOC emissions. The inventory for total VOC mass is
updated annually, whereas the speciation profiles are only periodically updated when new
information becomes available. Thus, trends in a particular species for a sector are a reflection of
changes in total VOC emissions for the sector and do not normally reflect any changes over time in
the speciation profile of the sector which may have occurred.
2.3 Photochemical box model
The framework for evaluating effects of changing VOCs speciation is a 0-D Atmospheric
Modelling chemistry box model (Wolfe et al., 2016), driven by the Version 3.3.1 of the Master
Chemical Mechanism (MCM v3.3.1) (Saunders et al., 2003; Jenkin et al., 2003). The model can be
effective in identifying the instantaneous *in- situ* $O_3$ sensitivity to changes in individual VOCs. The
measured concentrations of 38 VOCs species, $NO_x$, and CO, along with air temperature and pressure,
and relative humidity were averaged to a time resolution of 1-hour to constrain the model. A 3-day
model spin-up, with each 24-hour model run constrained by the observational data, was performed
in order to initialize the unmeasured compounds and transient radicals. The modelled outputs on the
4[th] day were taken as representing steady state of the photochemistry.
Photolysis rates were calculated as a function of solar zenith angle (Saunders et al., 2003):

$$J = l(\cos \chi)^{m} \exp(-n \sec \chi) \tag{1}$$

Where $J$ is the photolysis rate in $s^{-1}$; $l$, $m$, $n$ are constants derived from radiative transfer model
runs for clear sky condition at an altitude of 0.5 km and literature cross sections/quantum yields; $\chi$
is the solar zenith angle in radians.

The net production rate of $O_3$ ($P(O_3)$) is calculated by the difference of the production rate of

$O_3$ and the destruction rate of $O_3$, as in Equation (2):
$$
\begin{aligned}
P(O_3) = &\left(k_{HO_2+NO}[HO_2][NO] + \sum_i k_{RO_{2_i}+NO}[RO_2][NO]\right) - \\
&\left(k_{O^1D+H_2O}[O^1D][H_2O] + k_{O_3+OH}[O_3][OH] + k_{O_3+HO_2}[O_3][HO_2]\right. \\
&\left. k_{NO_2+OH}[NO_2][OH] + \sum_i k_{RO_{2_i}+NO_2}[RO_2][NO_2]\right)
\end{aligned}
\tag{2}
$$

Where the former part is the rate of $O_3$ production, representing by rate of NO oxidation by

$HO_2$ and $RO_2$ radicals; the latter part is the destruction rate of $O_3$, calculating by the sum of the rate
of $O_3$ photolysis, the rates of the reactions with OH and $HO_2$ radicals, and the rates of $NO_2$ loss
through reactions with OH and $RO_2$ radicals.

The sensitivity of $O_3$ to its precursors is quantified by the index of relative incremental

reactivity (RIR) (Liu et al., 2022b), as in Equation (3):
$$
RIR = \frac{\Delta P(O_3)}{P(O_3)} \times a^{-1}
\tag{3}
$$

Where RIR is the Relative Incremental Reactivity in %/%, $\Delta P(O_3)/P(O_3)$ is the ratio of the

change in $O_3$ production rate to the base $O_3$ production rate; $a$ is the reduction percentage in the
input concentration of $O_3$ precursors – a factor that allows for the effects of changing absolute
amounts of VOCs to be evaluated. Here a value of 30% was adopted for $a$.

## 3. Results and Discussion

3.1 Observation overview

The time series of $O_3$ and its precursors during August 2022 are shown in Figure 1, subdivided

into periods that will be referred to as 'initial period', '$O_3$ period', and 'clear–out'. The three periods
covered $1^{st}$ August-$21^{th}$ August 2022. Each period included one full week to avoid
weekday/weekend differences in $NO_x$ and VOCs concentrations impacting differently when $O_3$
production was compared between the three periods (de Foy et al., 2020). Ozone showed a generally
increasing trend from $1^{st}$ to $14^{th}$ August and then returned to relatively low concentrations after $15^{th}$
August 2022. The daily maximum 8 h average $O_3$ concentrations (MDA8h $O_3$) during the $O_3$ period
exceeded the WHO guideline value (100 μg m⁻³), ranging from 111 to 153 μg m⁻³. The elevated $O_3$
during the middle of the month corresponded to more intense photochemical formation under hot
weather conditions (32.7 ℃ in maximum) and higher concentrations of $O_3$ precursors (Table S3).

The diurnal profile of NO and $NO_2$ in the three periods generally showed a bimodal pattern,

albeit less pronounced in the initial and clear–out periods (Figure 2). The two peaks likely arise as
a consequence of increased traffic volumes at the start and end of the day, coupled to boundary layer
height changes in the early morning and into the evening (Lee et al., 2020). The average
concentrations of $NO_2$ during 05:00-10:00 were 10.8 ppbv in the $O_3$ period, which was considerably
higher than the concentration of 3.9 ppbv in the initial period and 3.4 ppbv in the clear–out period.
The low level of NO in the $O_3$ period highlights the rapid consumption of NO via photochemical
processes. The oxidation of CO is an important source of $HO_2$ in the atmosphere (Chen et al., 2020),
here in the range of 82.5 to 134.2 ppbv with little difference between periods. The diurnal profiles
of $O_3$ peaked at 15:00, with maximum hourly concentrations of 31.6, 67.2, and 30.4 ppbv in the
initial, $O_3$, and clear–out periods, respectively. Slight decreases in $O_3$ were observed during
nighttime (00:00-05:00), indicating enhanced NO titration effects.

The detailed VOCs composition in the three periods is presented in Figure S1. Concentration

of TVOC were 19.4 ± 8.4, 48.0 ± 18.8, and 23.5 ± 12.5 μg m$^{-3}$ in the three periods, respectively.
Alcohols, represented mainly by methanol and ethanol, were the predominant group that contributed
40.3% - 47.4% of over measured VOCs mass. This was followed by alkanes (21.4%-24.6%) and
ketones (16.3%-17.3%). Contributions of aldehyde (acetaldehyde), aromatics, alkenes, and
acetylene were low, ranging from 1.0% to 9.4% of total mass. Average mixing ratio of the top 10
species in selected periods at Birmingham Supersite are listed in Table S4. The top 10 species were
represented by methanol, acetone, ethanol, acetaldehyde, and $C_2 – C_4$ alkanes across initial period,
$O_3$ period, and clear-out period. The top individual species contributing to the total VOCs were
methanol (10.3% – 33.6%) and acetone (15.5% – 17.1%), regardless of the subdivided periods. The
results highlight large emissions of ethane, propane, n-butane, and i-butane associated with Natural
Gas (NG), Liquefied Petroleum Gas (LPG), and propellant use, fuel combustion and evaporation.
Ambient VOCs largely influenced by combustion-related sources (i.e., vehicle exhaust and coal
combustion) generally show alkane-dominated composition (Wu and Xie, 2017). Here, the
composition and amount of VOCs observed were most likely influenced by non-combustion
processes such as volatile chemical product usage and industrial processes (Gkatzelis et al., 2021).
Methanol was the most abundant VOC with an average concentration of 4.1 ppbv, followed by
acetone (2.0 ppbv), ethane (1.9 ppbv), ethanol (1.8 ppbv), and acetaldehyde (1.0 ppbv). The average
ratio of ethene/ethane was $0.2 \pm 0.1$ over the campaign, considerably lower than seen in polluted
locations, e.g. Hong Kong (China) ($0.7 \pm 0.1$) (Wang et al., 2018) and Seremban (Malaysia) (1.1)
(Zulkifli et al., 2022).

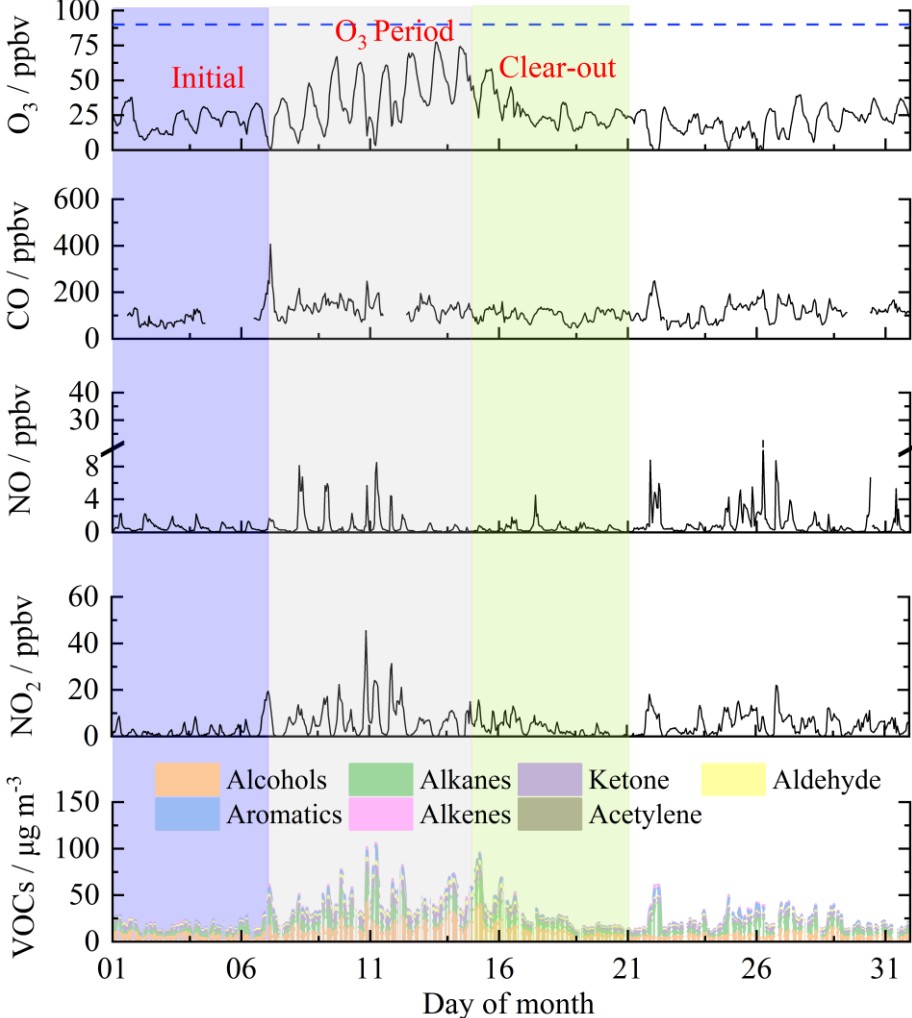


Figure 1. Time series of $O_3$, CO, NO, $NO_2$, and VOCs groups at the Birmingham Supersite. The
blue dash line denotes the national standard (90 ppbv) for hourly $O_3$ concentration.

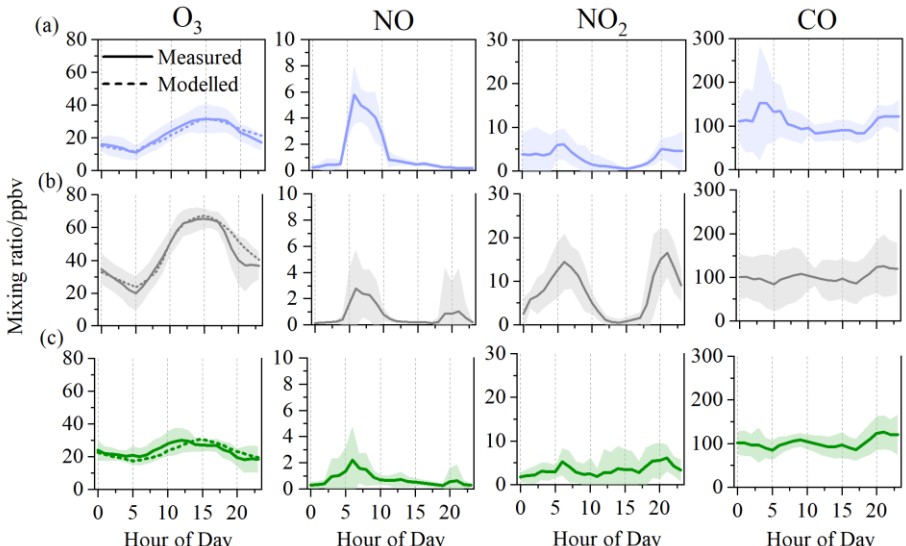

Figure 2. Diurnal variations of $O_3$, NO, $NO_2$, and CO during the initial (a), $O_3$ period (b), and clear–out period (c). The shaded areas represent standard variations.

The general diel profiles for all selected VOCs, except for ethane, showed bimodal pattern (Figure S2). Concentrations were much higher during the night, and lower in the day, due to they were subject to photochemical losses during the daytime. The bimodal pattern is less apparent for methanol and acetone, as they are abundant species originating from many anthropogenic sources in urban areas. For example, methanol was the most abundant species measured at a roadside in UK using Thermal Desorption-Gas Chromatography coupled with Flame Ionization Detection (TD-GC-FID) (Cliff et al., 2023). A separate study on gasoline and diesel vehicle exhausts reported methanol and acetone were the largest OVOCs emitted (Wang et al., 2022). Gkatzelis et al. (2021) conducted Positive Matrix factorization (PMF) analysis based on observed VOCs dataset in New York City, and concluded that acetone was the second most abundant species in measurements and was mostly attributed to volatile consumer product emissions (90%). (See Section 3.3 for further discussions for anthropogenic sources of OVOCs).

3.2 Observation-based $O_3$ formation sensitivity

The *in -situ* $O_3$ formation sensitivity was examined via reaction rates of ozone precursors and OH radical (OH reactivities, ($k$(OH))) and RIR scales of ozone precursors, along with the chemical budgets of $O_3$ formation and loss. In initial and clear-out periods. $k$(OH) exhibited consistent diurnal patterns, ranging from 2.4 to 5.9 s$^{-1}$ (Figure S3). In the $O_3$ period, $k$(OH) reached 9.0 and 8.7 s$^{-1}$ at approximately 07:00 and 20:00, respectively. A rapid increase in $k$(OH) was observed in the early

morning (00:00-06:00). VOCs and model generated species represented 60.5%, 65.7%, and 56.7% of the total $k$(OH) in the three periods, respectively. $NO_x$ and CO only contributed 10.2% -27.9% to total $k$(OH). Among of the VOCs groups, alcohols exhibited the largest $k$(OH) in all periods, accounting for 5.0% - 6.9% of the total $k$(OH). The diurnal production and loss of $O_3$ are shown in Figure S4. The oxidation and photolysis of VOCs promoted the production of $RO_2$, and $NO+RO_2$ contributed 47.7% of the $O_3$ production pathways in the $O_3$ period and 36.2% and 39.8% in initial and clear–out periods, respectively. Considering $O_3$ destruction, $OH+NO_2$ was the most important pathway during morning (08:00-12:00), accounting for 73.5%, 55.4% and 59.4% of the $O_3$ destruction pathways in the three periods. The dominant $OH+NO_2$ contribution to $O_3$ destruction suggested that the *in -situ* $O_3$ productions in all three periods was sensitive to VOCs emissions to some extent.

In order to understand contributions of $O_3$ formation from direct emissions and secondary formations of OVOCs, we developed two modelling scenarios: (1) all OVOCs species were constrained to observed mixing ratio; (2) all OVOCs species were unconstrained. (2) allowed secondary formations of OVOCs by oxidations of their precursor VOCs. As shown in Figure S5, secondary formations of OVOCs had little impact on $O_3$ formation in all periods. The simulation of $O_3$ production using the box model without constraining observed OVOCs slightly underestimated average daily maximum $O_3$ mixing ratio and P($O_3$), compared to the scenario with all observed OVOCs species constrained. The underestimation for average daily maximum mixing ratio of $O_3$ was 4.8%, 6.9%, and 5.1% in initial period, $O_3$ period, and clear-out period, respectively. In this case, the underestimation of average daily maximum P($O_3$) was 5.1%, 6.0%, and 9.3% in the three periods, respectively. The results demonstrated that in the Birmingham case study, primary emissions of OVOCs played central role in the *in -situ* ozone production.

The relative incremental reactivity of $NO_x$, CO, and anthropogenic VOCs (AVOCs, all measured VOCs except for isoprene) are shown in Figure 3. The *in -situ* $O_3$ production was most sensitive to anthropogenic VOCs with the highest positive RIR values (0.44 - 0.49). This is as anticipated given earlier analyses demonstrating their role in determining $k$(OH) and $O_3$ production. The low RIR (0.03 - 0.07) for CO in all three periods indicated a minor contribution of CO oxidation to $O_3$ production. The high RIR (0.24) for $NO_x$ was only observed in the $O_3$ period. Acetaldehyde showed the highest positive RIR (0.17 - 0.19) among the AVOCs, suggesting that the photolysis and

oxidation of acetaldehyde was a limiting factor for $O_3$ formation. The important role of carbonyl
compounds in atmospheric photochemistry has also been reported in previous studies, contributing
up to 59.3% to the $O_3$ formation in ambient environments in China, the United States, and Brazil
(Qin et al., 2021; Liu et al., 2022a; Edwards et al., 2014). Alkanes and alcohols exhibited lower RIR
values (0.02 - 0.04), despite their high mass concentrations.

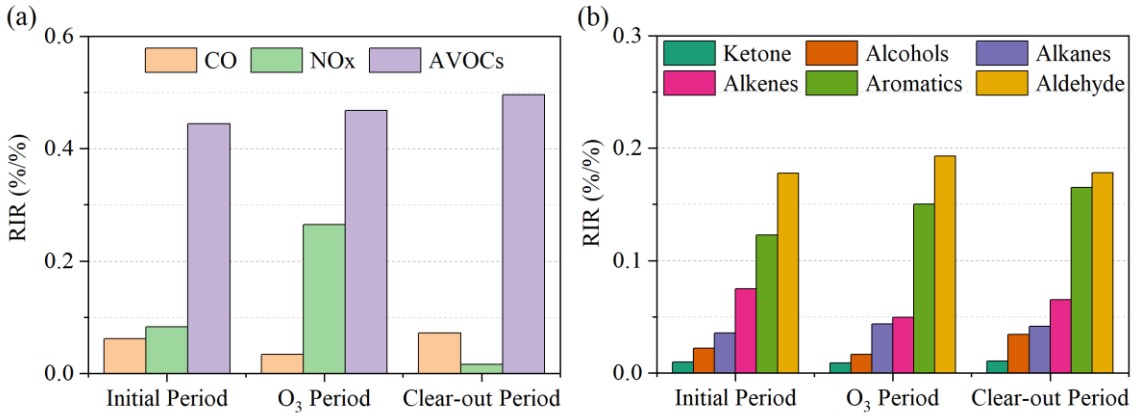


Figure 3. Modelled RIRs for (a) major $O_3$ precursors and (b) the AVOCs groups during
photochemically active daytime (08:00-16:00) in the selected periods. (AVOCs: anthropogenic
VOCs, all measured VOCs except for isoprene)

3.3 Emission inventory-informed $O_3$ production sensitivity tests
The trends in anthropogenic VOCs emissions from 1990 to 2019 estimated by the NAEI are
shown in Figure S6. Over the period, the annual national emissions decreased by ~69.0% from 2,941
kt in 1990 to 911kt in 2019. The reduction is partly attributed to more stringent controls for gasoline
vehicle emissions, both tailpipe and evaporative/fugitive. In 2019, VOCs emissions from on-road
transport and fuel fugitive losses accounted for only 3.3% and 13.7% of the total mass of VOCs
emissions, compared to 29.1% and 26.9% in 1990. Efforts have also been directed towards
controlling industrial processes, commercial solvent usage, and combustion emissions, resulting in
reductions of 66.8%, 48.9%, and 20.7%, respectively over the period. However, contributions from
solvent usage to total VOCs emissions over 1990-2019 showed only modest reductions in the 1990s
and 2000s and indeed small increases in the most recent years (Figure 4(b)). This slight growth in
solvent usage is due to increasing emissions from solvent use in consumer products such as
decorative products, aerosols, personal-care products, and detergents (NAEI, 2021). Solvent usage
had become the largest contributory sector (33.7%) to VOCs emissions by 2019, followed by
industrial processes (16.0%).

As shown in Figure 4(a), the VOC speciation over the 1990-2019 period was dominated in mass

terms by contributions from alkanes and alcohols, the former decreasing as gasoline sources
declined, the other increasing as non-industrial solvent and food and drink industry processes
emissions followed a different pattern. Alkane emissions fell from 46.6% to 30.6% over the period.
Further reductions in alkane emissions are expected from policies for that phase-out sales of new
internal combustion engine vehicles in the UK (and in many other places) by 2035. Growth in the
relative contributions of alcohols was primarily driven by increases in emissions of methanol and
ethanol, and to a lesser extent in 1-butanol and 2-propanol (Figure 4(c)).

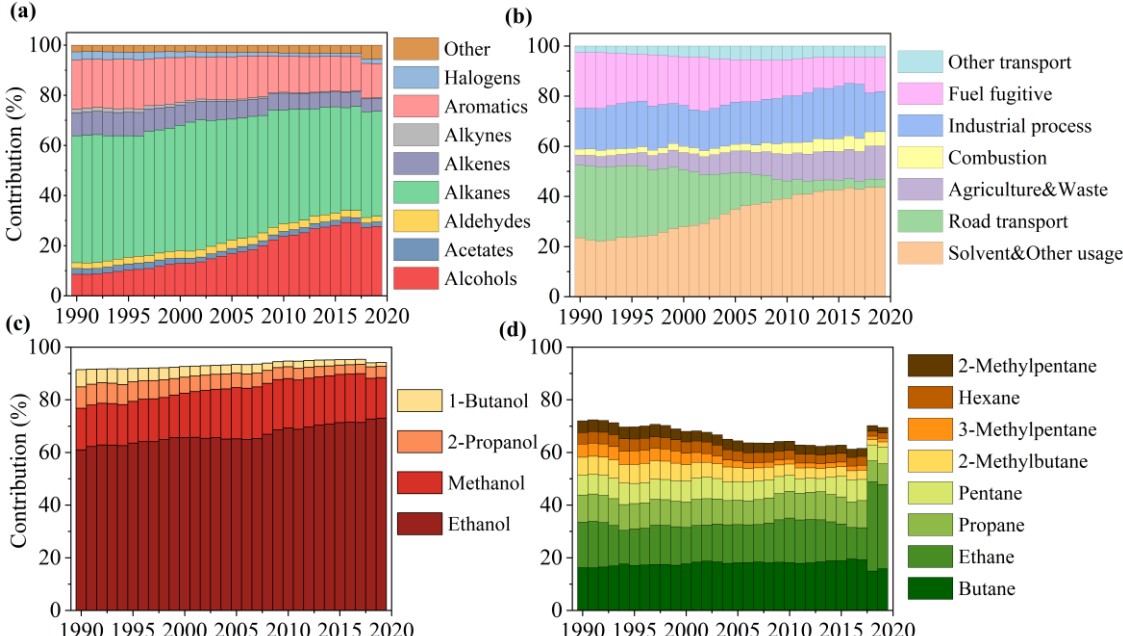


Figure 4. Contributions to annual national UK emissions of VOCs between 1990-2019 by: (a)
functional group; (b) by major emissions reporting sector; (c) for four individual alcohols in the
overall sub-class of alcohols; (d) for eight individual alkanes in the sub-class of all alkanes.

Figure S7 shows UK emission trends of individual species from different VOCs classes. These

highlight a national trend since 1990 of decreasing emissions of ethane associated with natural gas
leakage, toluene and propane associated with on-road petrol evaporation, as well as reductions of
benzene, ethene, and acetylene associated with tailpipe exhaust. Meanwhile, the reduced emissions
have been accompanied with increases in emissions of methanol and ethanol. The increase in
methanol is largely attributed to increased emissions from car-care products (i.e., non-aerosol
products). The increase in ethanol is due to increased domestic and industrial solvent usage.

The $NO_x$, CO, and VOC speciation within the NAEI for each of the six major emission sectors

was used to assign proportional sectoral contributions to the VOCs observed in Birmingham, and
hence to ozone production in the case study. (Table S5). Isoprene was excluded from the analysis as
it is assumed to have an entirely biogenic source. The six sectors are: road transport (both of on-
road exhaust emission and evaporative losses of fuel vapor), industrial processes, combustion,
solvent usage, fuel fugitive, and agriculture emissions). This makes a key assumption that the VOCs
at the observation site are affected directly in the same proportion that VOCs are reported in national
amounts in the NAEI. We make this assumption since it provides a reasonable starting point for
understanding how each VOC sector may influence $O_3$ production during a case study event,
however ozone formation might be sensitive to differing regional distribution in speciation. The
attribution of VOCs sources based on the NAEI data can be thought of as representative for this
case study as a typical urban environment, but it might not hold for cities near large industrial VOC
sources (i.e., oil refinery and industrial production sites), since they can significantly affect
composition and chemical reactivity of ambient VOCs.

Figure S8 shows the modelled RIRs for these sources in the initial, $O_3$, and clear–out periods.

All the sources generally showed higher RIR values in the $O_3$ period. Road transport exhibited the
highest positive RIR values in all periods (0.30 - 0.36), followed by industrial process (0.06 - 0.09)
and solvent usage (0.05 - 0.07). Despite being a relatively minor contributor to the mass of national
VOCs emissions (only 3.3% of the total in 2019), road transport VOCs still played the most
important role in local ozone photochemical chemistry, in this case study.

Figure 5a shows the changes in $P(O_3)$ during the $O_3$ period from 08:00 to 16:00 which might

arise as a result of reductions in the individual sectors described above. For this analysis, emission
changes in these sectors can be obtained by reducing model constrained concentrations of VOCs,
$NO_x$, and CO according to their contributions arising from individual emission sectors. This is a
'thought experiment' where under 2019 general observed atmospheric conditions (e.g., for $NO_x$, CO
and so on), each of the VOC source sectors is then further reduced in isolation (from 2019 levels)
and the effects on $\Delta P(O_3)$ were evaluated. Based on these scenarios, reducing emissions from the
individual sectors all resulted in decreased $P(O_3)$, as would be anticipated. Reducing ozone
precursors arising from road transport would lead to a decreased $P(O_3)$ of ~1.71 ppbv $h^{-1}$ if that

sector could be 100% abated in the case study. This is expected because road transport is a source

of photochemically reactive VOCs, including aromatics, aldehyde, and short-chain alkanes/alkenes.

Other sectors showed more modest effects, with reductions in solvent-related VOCs the next most

significant lever to control ozone. Fully abating emissions of all industrial and solvent process

emissions only resulted in a decreased $P(O_3)$ of ~0.35 ppbv h$^{-1}$, largely because they are dominated

by ethanol and methanol with relatively low RIR values. Considering the real-world changes in

VOC emissions over the period of 1990 to 2019, the very major reductions in road transport

emissions have led to the largest effects in reducing $P(O_3)$ (Figure 5(b)). Whilst there have also been

some very large reductions (94.6%) in fuel fugitive emissions, the impact on $P(O_3)$ reduction is

modelled to have been relatively modest, being similar to industrial processes and solvent usage. It

is important to acknowledge the limitation of this analysis. In the real world, reductions in ambient

VOCs from the NAEI sectors are affected by photochemical loss rate and advection processes,

potentially altering the proportion of VOCs that would be observed with each sector reduction at

the measurement site. This would potentially be an important consideration if instantaneously

radical budgets were being evaluated, but it is a less significant issue when integrated ozone

production effects.

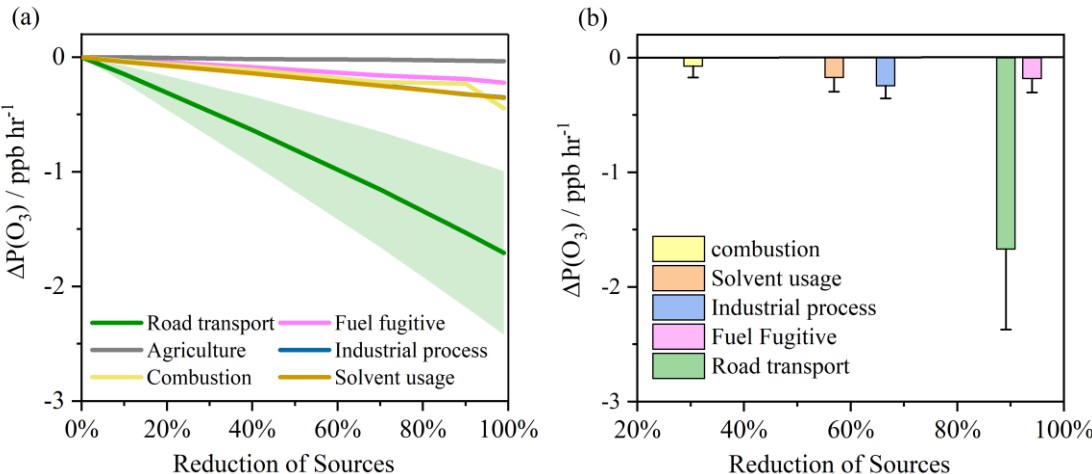

Figure 5. (a) Changes in $P(O_3)$ in response to different reductions in VOCs, NO$_x$, and CO from

different sectors for the Birmingham-case study condition. (b) Changes in $P(O_3)$ based on the NAEI

estimated reductions in VOCs from different sectors between 1990 and 2019. The standard

deviations represent variability in $\Delta P(O_3)$ during 08:00-16:00 LST in the O$_3$ period.

Further model runs were performed to better understand the impacts of the shift between

alkanes and alcohol species on P(O$_3$), given trends showing decreasing alkanes emissions and
increasing alcohol emissions between the 1990-2019 period (Figure 4). The modelled alkane
concentrations in the case study were reduced by 10%, 30%, 50%, 70%, 90%, and 99%. This
represents a downward trajectory in alkane emissions that would be anticipated as gasoline vehicles
are slowly retired. Two further scenarios were then developed to sit alongside these reductions in
alkanes. Firstly, the concentration of ethanol was increased to keep the overall total VOC
concentrations in the model under case study conditions unchanged. Second, the concentration of
both ethanol and methanol were scaled upwards to keep total VOCs concentration unchanged. As
shown in Figure 6, reductions in alkanes alone resulted in decreased P(O$_3$) to a maximum of ~0.26
ppbv h$^{-1}$ if fully abated. If that alkane reduction was balanced with increased ethanol and methanol,
then $\Delta$P(O$_3$) is reduced by a maximum of 0.19 ppbv h$^{-1}$. If alkane reductions were balanced by
increasing ethanol alone, then P(O$_3$) still decreases, but only up to 0.07 ppbv h$^{-1}$.

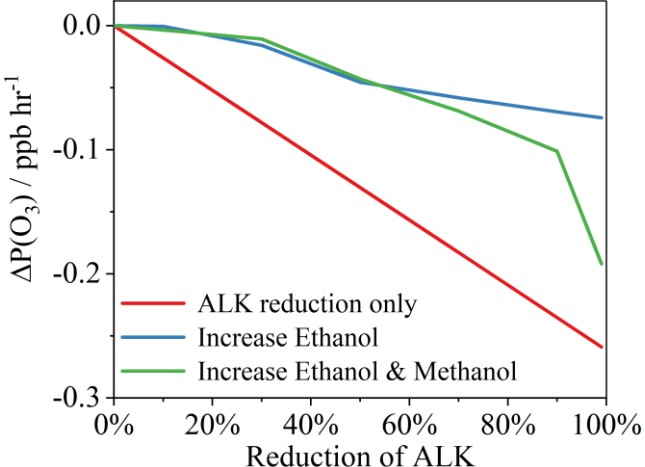


Figure 6. Reductions in $\Delta$P(O$_3$) based on reducing alkanes (ALK) in the model (under case study
conditions), reducing alkanes but balancing the overall VOC amount with increased ethanol (blue
line) and reducing alkanes but balancing the overall VOC amount with increased ethanol and
methanol (green line).

## Conclusion

In this study a typical high-O$_3$ event in Birmingham, United Kingdom was chosen as a case
study to investigate the impacts of changes to VOCs emissions and speciation on urban O$_3$
production. The *in- situ* O$_3$ formation sensitivity was split into three periods: initial, high O$_3$, and
clear–out. Results from OH reactivity, $O_3$ budgets, and RIR index showed that $O_3$ formation in all
three periods was impacted by both VOCs and $NO_x$, but was more sensitive to anthropogenic VOCs.
The oxidation of alcohols and photolysis of acetaldehyde substantially contributed to *in- situ* $O_3$
formation, especially in the high $O_3$ period. The roles of anthropogenic VOC sources in urban $O_3$
chemistry were examined by integrating the NAEI speciation over the period of 1990-2019 into
photochemical box model scenarios. Despite road transport only contributing 3.3% of national
VOCs emissions in 2019 it still played the most important VOC role in the case study ozone
photochemistry, when inventory contributions were mapped onto observed VOCs. Sequentially the
observed VOCs were reduced by the fractional contributions and speciation in the NAEI for six
sectors to evaluate what impact abating different VOCs-emitting sectors would have on $P(O_3)$.
Abating road transport VOCs in isolation would lead to a decreased $P(O_3)$ by up to 1.67 ppbv h$^{-1}$,
but abating other sectors such as solvent use and fugitive fuels had noticeably smaller effects.
Despite emissions of VOCs from road transport falling very dramatically between 1990 and 2019,
it remains one of the most powerful means to further reduce ozone in this typical UK case study.
The wider shift in speciation reported in the NAEI from alkanes to alcohols was also examined
using scenarios where emission reductions for alkanes, were counterbalanced with increases in
alcohols, all simulated for the Birmingham case study conditions (e.g., for $NO_x$, CO and etc). Further
reducing alkanes from present day conditions to zero has a clear beneficial effect on reducing $P(O_3)$
by up to ~0.26 ppb hr$^{-1}$. However, this benefit would to a degree be offset should alcohol emissions
(for example from food and drink, and/or solvent use) increase to counterbalance those alkanes
reductions. Whilst simple alcohols are inherently less potent ozone-forming VOCs compared to the
mixture of VOCs from road transport, avoiding future growth in emissions remains important, since
they weaken the long-term benefits of road transport electrification and the phase out of internal
combustion engine vehicles.

## Data Availability

Observational data including meteorological parameters and air pollutants used in this study are
available at https://github.com/nervouslee/Birmingham_CS.git. UK national emission inventory is
available at https://naei.beis.gov.uk/.

## Author Contribution

Jianghao Li prepared the manuscript with contributions from all authors. Alastair C. Lewis helped with modelling scenarios and revised the manuscript. Jim R. Hopkins contributed to measurement of chemical species. Stephen J. Andrews contributed to scientific discussion on findings of this work. Tim Murrells, Neil Passant and Ben Richmond contributed to the data of national emission inventory data and revision on NAEI methodology. Siqi Hou, William J. Bloss, Roy M. Harrison, and Zongbo Shi provided measurements of atmospheric pollutants used in this study, along with critical discussion on revising the manuscript.

## Competing interests

The authors declare that they have no conflict of interest.

## Acknowledgements

Establishment and operation of the Birmingham Air Quality Supersite operation (BAQS) is supported by the NERC WM-Air project (NE/S003487/1) and UKRI Clean Air SPF project OSCA (NE/T001976/1). This work forms part of the National Centre for Atmospheric Science National Capability programme funded by NERC. Jianghao Li's study at the University of York is financially supported by the China Scholarship Council (grant no. 202206560052).

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
