# Peer review of "The impact of multi-decadal changes in VOCs speciation on urban"

_EGUsphere, 2023_

## Author Comment (AC1)

**Reviewer #1**

I think that the presentation quality requires substantial improvements in three aspects, as follows.

1) The VOC observed datasets are poorly presented. This manuscript did not thoroughly discuss the trend of speciated VOCs but lumped the species into functional groups. It is impossible to evaluate the model evaluation processes without VOC-specific information. In addition, FID is not a common analytical tool to quantify oxygenated VOCs, which requires a thorough description of the analytical characteristics;

**Authors' reply:** We very much appreciate the time and effort you have put into your comments. Your advice about completing descriptions on VOCs observed datasets is most helpful for improving the quality of this manuscript. We have now added figures and manuscript texts to increase the information on important VOCs species. Please see specific changes as follows:

■ Campaign averaged mixing ratio of the individual VOCs species have been added in supplement material, as shown below:

Table 1. Average mixing ratio in ppbv and effective carbon number (ECN) of the measured VOCs at Birmingham Supersite over August, 2022.

|  | Species | Mean | SD | ECN |
|---|---|---|---|---|
| **Alkanes** | ethane | 1.69 | 1.48 | – |
|  | propane | 0.65 | 0.57 | – |
|  | i-butane | 0.27 | 0.27 | – |
|  | n-butane | 0.50 | 0.46 | – |
|  | cyclopentane | 0.03 | 0.04 | 5.00 |
|  | i-pentane | 0.19 | 0.17 | – |
|  | n-pentane | 0.09 | 0.10 | – |
|  | 2-methylpentane | 0.05 | 0.05 | – |
|  | 3-methylpentane | 0.03 | 0.03 | 5.00 |
|  | hexane | 0.03 | 0.03 | – |
|  | heptane | 0.02 | 0.02 | – |
|  | i-octane | 0.02 | 0.02 | – |
|  | nonane | 0.06 | 0.03 | 9.00 |
| **Alkenes** | ethene | 0.26 | 0.20 | – |
|  | propene | 0.10 | 0.08 | – |
|  | t-2-butene | 0.01 | 0.01 | – |
|  | 1-butene | 0.02 | 0.02 | – |
|  | i-butene | 0.02 | 0.01 | 4.00 |
|  | c-2-butene | 0.00 | 0.00 | – |
|  | 1,3-butadiene | 0.01 | 0.01 | – |
|  | t-2-pentene | 0.00 | 0.01 | – |
|  | c-2-pentene | 0.01 | 0.01 | – |
|  | isoprene | 0.12 | 0.13 | – |
| **Alkyne** | acetylene | 0.11 | 0.06 | – |
| **Aromatics** | benzene | 0.07 | 0.05 | – |
|  | toluene | 0.16 | 0.14 | – |
|  | ethylbenzene | 0.04 | 0.04 | – |
|  | m-xylene | 0.11 | 0.12 | – |
|  | p-xylene | 0.04 | 0.04 | – |
|  | o-xylene | 0.04 | 0.05 | – |
|  | 1,3,5-trimethoxybenzene | 0.01 | 0.01 | – |
|  | 1,2,4-trimethoxybenzene | 0.05 | 0.06 | – |
|  | 1,2,3-trimethoxybenzene | 0.01 | 0.02 | – |
| **OVOCs** | acetaldehyde | 1.09 | 0.59 | 1.00 |
|  | acetone | 2.21 | 1.13 | 2.00 |
|  | methanol | 3.72 | 2.35 | 0.75 |
|  | ethanol | 1.79 | 1.60 | 1.50 |

- A table that summarizes mean mixing ratio of the top 10 species in initial period, $O_3$ period, and clear-out period have been added into supplement material, as shown below:

**Table 2.** Average mixing ratio (ppbv) of the top 10 species in selected periods at Birmingham Supersite.

| Ranking | Initial period | | $O_3$ period | | Clear-out period | |
|---|---|---|---|---|---|---|
| | Species | Concentration | Species | Concentration | Species | Concentration |
| 1 | methanol | $2.60 \pm 0.90$ | methanol | $6.45 \pm 2.03$ | methanol | $3.92 \pm 1.82$ |
| 2 | acetone | $1.66 \pm 0.53$ | acetone | $3.90 \pm 0.80$ | acetone | $1.80 \pm 0.94$ |
| 3 | ethane | $1.43 \pm 1.40$ | ethanol | $3.33 \pm 2.27$ | ethanol | $1.42 \pm 0.01$ |
| 4 | ethanol | $1.40 \pm 1.08$ | ethane | $2.32 \pm 1.79$ | ethane | $1.22 \pm 0.88$ |
| 5 | acetaldehyde | $0.85 \pm 0.31$ | acetaldehyde | $2.00 \pm 0.38$ | acetaldehyde | $0.82 \pm 0.41$ |
| 6 | propane | $0.48 \pm 0.37$ | propane | $1.05 \pm 0.84$ | propane | $0.49 \pm 0.38$ |
| 7 | n-butane | $0.37 \pm 0.31$ | n-butane | $0.75 \pm 0.57$ | n-butane | $0.35 \pm 0.31$ |
| 8 | ethene | $0.21 \pm 0.17$ | i-butane | $0.39 \pm 0.32$ | i-butane | $0.24 \pm 0.33$ |
| 9 | i-butane | $0.19 \pm 0.16$ | ethene | $0.35 \pm 0.20$ | ethene | $0.19 \pm 0.14$ |
| 10 | isoprene | $0.13 \pm 0.12$ | i-pentane | $0.32 \pm 0.23$ | i-pentane | $0.13 \pm 0.11$ |

Average mixing ratios of the top 10 species in selected periods at Birmingham Supersite are listed in Table 2. The top 10 species were represented by methanol, acetone, ethanol, acetaldehyde, and $C_2 – C_4$ alkanes across initial period, $O_3$ period, and clear-out period. The top individual species contributing to the total VOCs were methanol (10.3% – 33.6%) and acetone (15.5% – 17.1%), regardless of the subdivided periods. The results highlight large emissions of ethane, propane, n-butane, and i-butane associated with Natural Gas (NG), Liquefied Petroleum Gas (LPG), and propellant use, fuel combustion and evaporation. Above description has been added into the revised manuscript **Page 9 Line 254-260**.

- A figure of diurnal variations of ethene, ethanol, toluene, methanol, ethane, acetylene, acetaldehyde, and acetone in the selected period have been added into supplement material, as shown below:

[Figure]

**Figure 1.** Campaign averaged diel mixing ratio of selected VOCs during different periods: ethene, ethanol, toluene, methanol, ethane, acetylene, acetaldehyde, and acetone. The shaded area represents one standard deviation from the mean.

The general diel profiles for all selected VOCs, except for ethane, showed bimodal pattern (Figure 1). Concentrations were much higher during the night, and lower in the day, due to they were subject to photochemical losses during the daytime. The bimodal pattern is less apparent for methanol and acetone, as they are abundant species originating from many anthropogenic sources in urban areas. For example, methanol was the most abundant species measured at a roadside in UK using Thermal Desorption-Gas Chromatography coupled with Flame Ionization Detection (TD-GC-FID) (Cliff et al., 2023). A separate study on gasoline and diesel vehicle exhausts reported methanol and acetone were the largest OVOCs emitted (Wang et al., 2022). Gkatzelis et al. (2021) conducted Positive Matrix factorization (PMF) analysis based on observed VOCs dataset in New York City, and concluded that acetone was the second

most abundant species in measurements and was most attributed to volatile consumer product emissions (90%). (See Section 3.3 for further discussions for anthropogenic sources of OVOCs). Above texts have been added into the revised manuscript **Page 11 Line 278-289**.

- As for your comment "FID is not a common analytical tool to quantify oxygenated VOCs, which requires a thorough description of the analytical characteristics," the method being used here has been in use, and reported in literature, for more than 20 years – see: Hopkins, J. R., Lewis, A. C., and Read, K. A.: A two-column method for long-term monitoring of non-methane hydrocarbons (NMHCs) and oxygenated volatile organic. The method has been used in support of more than 40 publications by the York research group over the years. The challenges of calibration are no different to MS detection. Here we use direct calibration using 4 ppbv gas standard cylinders and the use of carbon response for this instrument is tightly controlled, with regular carbon-wise responses calculated for all species. Detection limits for this instrument are not higher than 0.1 ppbv and are typically in the 5-10 pptv range. Table 3 lists which species were directly calibrated, and which used equivalent carbon numbers for quantification. Additionally, please see Table 1 for effective carbon numbers of species which used carbon-wise responses. These tables have now been added into the revised supplement. Calibration sequence has been added into the revised manuscript **Page 6 Line 151-156**.

**Table 3.** Species quantified and their corresponding quantification method used in this study.

| Species | Quantification method |
| --- | --- |
| ethane | 4 ppbv gas standard cylinders |
| propane | 4 ppbv gas standard cylinders |
| i-butane | 4 ppbv gas standard cylinders |
| n-butane | 4 ppbv gas standard cylinders |
| i-pentane | 4 ppbv gas standard cylinders |
| n-pentane | 4 ppbv gas standard cylinders |
| 2-methylpentane | 4 ppbv gas standard cylinders |
| hexane | 4 ppbv gas standard cylinders |
| heptane | 4 ppbv gas standard cylinders |
| i-octane | 4 ppbv gas standard cylinders |
| ethene | 4 ppbv gas standard cylinders |
| propene | 4 ppbv gas standard cylinders |
| t-2-butene | 4 ppbv gas standard cylinders |
| 1-butene | 4 ppbv gas standard cylinders |
| c-2-butene | 4 ppbv gas standard cylinders |
| 1,3-butadiene | 4 ppbv gas standard cylinders |
| t-2-pentene | 4 ppbv gas standard cylinders |
| c-2-pentene | 4 ppbv gas standard cylinders |
| isoprene | 4 ppbv gas standard cylinders |
| acetylene | 4 ppbv gas standard cylinders |
| benzene | 4 ppbv gas standard cylinders |
| toluene | 4 ppbv gas standard cylinders |
| ethylbenzene | 4 ppbv gas standard cylinders |
| m-xylene | 4 ppbv gas standard cylinders |
| p-xylene | 4 ppbv gas standard cylinders |
| o-xylene | 4 ppbv gas standard cylinders |
| 1,3,5-trimethoxybenzene | 4 ppbv gas standard cylinders |
| 1,2,4-trimethoxybenzene | 4 ppbv gas standard cylinders |
| 1,2,3-trimethoxybenzene | 4 ppbv gas standard cylinders |
| cyclopentane | effective carbon number using toluene as reference |
| 3-methylpentane | effective carbon number using toluene as reference |
| nonane | effective carbon number using toluene as reference |
| i-butene | effective carbon number using toluene as reference |
| acetaldehyde | effective carbon number using toluene as reference |
| acetone | effective carbon number using toluene as reference |
| methanol | effective carbon number using toluene as reference |
| ethanol | effective carbon number using toluene as reference |

**References**

Cliff, S. J., Lewis, A. C., Shaw, M. D., Lee, J. D., Flynn, M., Andrews, S. J., Hopkins, J. R., Purvis, R. M., and Yeoman, A. M.: Unreported VOC emissions from road transport including from electric vehicles, Environmental science & technology, 10.1021/acs.est.3c00845, 2023.

Gkatzelis, G. I., Coggon, M. M., McDonald, B. C., Peischl, J., Gilman, J. B., Aikin, K. C., Robinson, M. A., Canonaco, F., Prevot, A. S., and Trainer, M.: Observations confirm that volatile chemical products are a major source of petrochemical emissions in US cities, Environmental science & technology, 55, 4332-4343, 10.1021/acs.est.0c05471, 2021.

Hopkins, J. R., Lewis, A. C., and Read, K. A.: A two-column method for long-term monitoring of non-methane hydrocarbons (NMHCs) and oxygenated volatile organic compounds (o-VOCs), Journal of Environmental Monitoring, 5, 8-13, 10.1039/B202798D, 2003.

Wang, S., Yuan, B., Wu, C., Wang, C., Li, T., He, X., Huangfu, Y., Qi, J., Li, X.-B., and Sha, Q. e.: Oxygenated volatile organic compounds (VOCs) as significant but varied contributors to VOC emissions from vehicles, Atmospheric Chemistry and Physics, 22, 9703-9720, 10.5194/acp-22-9703-2022, 2022.

2)  The emission inventory's speciation information is not thorough enough. All inventory information is presented in a lumped fashion except for alkane. It is vague how this information was integrated into the box model framework;

**Authors' reply:** The National Atmospheric Emission Inventory (NAEI) estimates emissions of over 600 different VOCs from anthropogenic sources. Due to the large number of VOCs species, we showed the decadal changes (1990 – 2019) of VOCs emissions from the perspective of the source sectors and VOCs classes, rather than attempting to represent such a large range of individual species.

**From the perspective of the source sectors:** First, decadal changes in 9 emission sectors were presented (Figure 2). Second, all observed VOCs (excluding isoprene) were attributed to the 2019 emission inventory sectors (Table 4). This makes a key assumption that the VOCs, $NO_x$, and CO at the observation site are affected directly in the same proportion that VOCs are reported in national amounts in the NAEI. Then, corresponding relative contributions of every VOCs species, $NO_x$, and

CO emitted from the six emission inventory sectors were integrated into the photochemical box model. In this sense, whilst species are bulked together for presentation purposes, they are treated explicitly in the chemical modeling. The ozone sensitivity of those sources was explicitly examined under different scenarios.

[Figure]

**Figure 2.** Emissions of VOCs from anthropogenic sources in the UK between 1990-2019. Data: UK National Atmospheric Emissions Inventory (https://naei.beis.gov.uk/, last access 07 September 2023).

**Table 4.** Relative contributions (%) of ozone precursors emitted from the six emission inventory sectors.

| | road transport | fuel fugitive | agriculture | industrial process | combustion | solvents | SUM |
|---|---|---|---|---|---|---|---|
| **ethane** | 6.0 | 48.6 | 39.1 | 2.1 | 2.9 | 0.0 | 98.7 |
| **butanes** | 34.9 | 35.2 | 0.0 | 1.4 | 1.2 | 27.0 | 99.8 |
| **propanes** | 82.0 | 9.9 | 0.0 | 0.7 | 0.5 | 6.7 | 99.8 |
| **C>=6 alkanes** | 39.5 | 31.3 | 0.0 | 2.2 | 1.5 | 22.8 | 97.2 |
| **acetylene** | 85.8 | 7.6 | – | 2.7 | 0.0 | – | 96.2 |
| **ethene** | 8.6 | 86.9 | – | 4.5 | – | – | 100.0 |
| **butenes** | 96.1 | 0.7 | – | 0.7 | 1.5 | – | 99.0 |
| **propene** | 64.1 | 34.1 | – | 1.8 | – | – | 100.0 |
| **pentenes** | 100.0 | – | – | – | – | – | 100.0 |
| **1,3-butadiene** | 76.0 | 3.3 | – | 3.5 | 11.0 | – | 93.8 |
| **toluene** | 80.0 | 3.8 | 0.3 | 0.6 | 1.1 | 10.3 | 96.1 |
| **xylenes** | 72.0 | 1.3 | 0.3 | 1.0 | 1.3 | 21.6 | 97.6 |
| **other aromatics** | 71.3 | 2.9 | – | 1.8 | 5.3 | 12.6 | 94.0 |
| **acetaldehyde** | 69.0 | – | 0.2 | 13.0 | 0.0 | – | 82.1 |
| **acetone** | 17.0 | – | – | 15.4 | 0.2 | 65.6 | 98.3 |
| **methanol** | – | 0.0 | – | 3.0 | – | 96.8 | 99.8 |
| **ethanol** | 7.3 | 0.1 | 11.9 | 48.8 | 5.8 | 25.3 | 99.1 |
| **NO$_x$** | 33.3 | – | 3.9 | 18.4 | 28.0 | – | 83.5 |
| **CO** | 14.5 | 1.2 | – | 32.3 | 34.0 | – | 82.0 |

**From the perspective of the VOCs classes**: The specaiation changes are shown in Figrue 3. Results highlighted dominant contributions from alkanes and alcohols in mass terms, the former decreasing as gasoline sources declined, the other increasing as non-industrial solvent and food and drink industry processes emissions followed a different pattern (Figure 3 (a), (b)). We then emphasized the roel of key individual species from alkanes and alcohols (Figure 3 (c), (d)). Contributions of methanol and ethanol to the total alcohol emissions were high over 1990 – 2019 period, increasing from 76.8% in 1990 to 88.4% in 2019. In order to understand the role of the speciation changes in ozone chemistry, we developed a scenario that decreased the mixing ratio of all observed alkanes (this covers the 7 major species in the emission inventory for alkanes) and increased the mixing ratio of methanol and ethanol since they represent most of the national alcohol group emissions. The scenario was further integrated into the box model, and corrseponding changes in ozone production rate was obtained.

[Figure]

**Figure 3.** Contributions to annual national UK emissions of VOCs between 1990-2019 by: (a) functional group; (b) by major emissions reporting sector; (c) for four individual alcohols in the overall sub-class of alcohols; (d) for eight individual alkanes in the sub-class of all alkanes.

■ We hope that above explanation helps understand how speciation changes are integrated into the box model in this study. It is inevitably a challenge to produce a concise representation of

sectors and emissions, when the underlying VOC complexity is very high. We hope this assures the reviewer that individual VOC species effects are properly included in the modelling and later conclusions. In responding to the concerns on emission inventory's speciation information, Figure 4 describes UK emission trends of individual species from different VOCs classes. This has been added into the revised supplement material. The result highlights a national trend since 1990 of decreasing emissions of ethane associated with natural gas leakage, toluene and propane associated with on-road petrol evaporation, as well as reductions of benzene, ethene, and acetylene associated with tailpipe exhaust. Meanwhile, the reduced emissions have been accompanied with increases in emissions of methanol and ethanol. The increase in methanol is largely attributed to increased emissions from car-care products (i.e., non-aerosol products). The increase in ethanol is due to increased domestic and industrial solvent usage. The above text has been added into revised manuscript **Page 14-15 Line 364-370**. Additionally, please see our previous publication for detailed information on NAEI estimated UK emissions of VOCs over 1990 – 2017 (Lewis et al., 2020).

[Figure]

**Figure 4.** Estimated trends (1990 – 2019) in the UK emissions of selected species corresponding to different VOCs classes.

**References**

Lewis, A. C., Hopkins, J. R., Carslaw, D. C., Hamilton, J. F., Nelson, B. S., Stewart, G., Dernie, J., Passant, N., and Murrells, T.: An increasing role for solvent emissions and implications for future

measurements of volatile organic compounds, Philosophical Transactions of the Royal Society A, 378, 20190328, 10.1098/rsta.2019.0328, 2020.

3) The sensitivity tests of the box model need to be conducted and thoroughly discussed. The oxidation product accumulation in the box model frame must be verified by comparing it with the observed value. This is particularly important as this study concludes that OVOCs play an important role in ozone production. Therefore, it is important to present a quantitative discussion of how much of OVOCs in the studied area is coming from direct emission vs photochemical production.

**Authors' Reply:** This is a good point to raise and we have adapted the manuscript in response. The reactivity of OVOCs arising through secondary formation at the Birmingham Supersite is a key issue when simulating impacts of changes in primary emissions of OVOCs on production rate of $O_3$ ($P(O_3)$). The text below and figures show results from the two scenarios are now included in the revised manuscript **Page 12 Line 307-318** and supplement material.

■ In order to understand contributions of $O_3$ formation from direct emissions and secondary formations of OVOCs, we developed two modelling scenarios: (1) all OVOCs species were constrained to observed mixing ratio; (2) all OVOCs species were unconstrained. (2) allowed secondary formations of OVOCs by oxidations of their precursor VOCs. As shown in Figure 5, secondary formations of OVOCs had little impact on $O_3$ formation in all periods. The simulation of $O_3$ production using the box model without constraining observed OVOCs slightly underestimated average daily maximum $O_3$ mixing ratio and $P(O_3)$, compared to the scenario with all observed OVOCs species constrained. The underestimation for average daily maximum mixing ratio of $O_3$ was 4.8%, 6.9%, and 5.1% in initial period, $O_3$ period, and clear-out period, respectively. In this case, the underestimation of average daily maximum $P(O_3)$ was 5.1%, 6.0%, and 9.3% in the three periods, respectively. The results demonstrated that in the Birmingham case study, primary emissions of OVOCs played central role in the *in -situ* ozone production.

[Figure]

**Figure 5.** Modelled O$_3$ mixing ratio (a, b, c) and P(O$_3$) (d, e, f) with and without photodegradable OVOCs during the select periods.

---

## Author Comment (AC2)

**Reviewer #2**

The manuscript entitled " The impact multi-decadal of changes in VOCs speciation on urban ozone chemistry: A case study in Birmingham, United Kingdom" aimed to quantify the impacts of the real-world changes in VOCs sources on urban $O_3$ production rate, and also evaluate the relative importance of different VOCs functional group classes on the $O_3$ production. The manuscript provides some valuable advice for the pollution control of $O_3$ in the area with relative clean air quality. I therefore suggest a necessary revision of this manuscript before final publication in Atmospheric Chemistry and Physics.

**Authors' reply:** We appreciate the reviewer's efforts in recognizing the contribution of our results to the research field of urban ozone pollution. We have been able to incorporate changes in responding to the reviewer's valuable feedbacks. Revised texts within the manuscript have been marked by red color, and newly added Table/Figure captions have been highlighted by yellow. Here is a point-by point response to the reviewer's comments.

1) The innovation of the paper needs further unearthing, as there are already many similar literatures;

**Authors' reply:** We value the reviewer's efforts in comparing our study to those in the existing literatures. Throughout the revised Introduction, we have enriched literature reviews (**Page 3, Line 60-62, Line 81-85** in revised manuscript) on previous $O_3$ studies. The current research gap has been clearly stated (**Page 3-4, Line 88-92** in revised manuscript), and the significance of our results has been pointed out (**Page 4, Line 102-104** in revised manuscript). The innovation of this paper is briefly denoted as follows:

■ Over the last decades, there have been extensive studies investigating controlling factors of *in -situ* production of $O_3$ from the perspective of chemical control regime and reactivity of $O_3$ precursors (please see detailed literature review on **Page 2-3 Line 47-63** in revised manuscript). For example, a recent study in a coastal city of China applied observation-constrained box model to clarify sensitivity of $O_3$ production and OH reactivity of VOCs classes during a high $O_3$ event in 2019 autumn. The results indicated that the $O_3$ production at

the urban site was VOC-sensitive. Aromatics, alkenes, and alkanes were the primary reduction target for the ozone pollution control, which showed the highest OH reactivity and played a leading role in radical recycling and $O_3$ production (Liu et al., 2022).

Since VOC emissions are often the limiting factor in photochemical production of urban ozone, the issue of shifts in major VOC emissions (detailed discussion please see **Page 3 Line 64-88** in revised manuscript) and their resulting impacts on urban ozone chemistry have been addressed worldwide. The increasing role for volatile chemical product (VCP) emissions in urban ozone chemistry has been taken into account. In North America and Europe cities, OVOCs emitted from volatile chemical products (VCP) can outweigh fossil fuel sources for urban VOCs. Modelling results showed that the additional OVOCs from VCP emissions were the most important species for urban $O_3$ production, increasing the daily maximum $O_3$ mixing ratio by as much as 10 ppbv in Los Angeles and 11 ppbv in New York (Coggon et al., 2021; Qin et al., 2021).

Although investigations on *in-situ* urban ozone chemistry and attributions of $O_3$ production from important VOC sources have been extensively conducted in atmospheric modeling studies, there has been limited reporting on the evaluation of real-world emission changes in VOCs speciation. From the perspective of *in-situ* $O_3$ production, the benefit of substantial reductions on vehicle emissions, whilst there has been a parallel increasing role for non-industrial solvent usage remains unclear. What effect this shift in speciation is having on ozone chemistry is less well studied. Therefore, by incorporating the detailed NAEI VOCs emission inventories over the period of 1990-2019 into a box model, $O_3$ formation in Birmingham is used as a case study to quantify the impacts of the real-world changes in VOCs sources on urban $O_3$ production rate. The study makes a major advance since it couples both highly detailed *in- situ* measurements along with a multi-year highly speciated inventory, something that has not been available to other studies. The results help understand impacts of decades of abating different VOCs-emitting sectors on urban $O_3$ production, and outline the implications for future $O_3$ control strategies.

**References**

Coggon, M. M., Gkatzelis, G. I., McDonald, B. C., Gilman, J. B., Schwantes, R. H., Abuhassan, N.,

Aikin, K. C., Arend, M. F., Berkoff, T. A., Brown, S. S., Campos, T. L., Dickerson, R. R., Gronoff, G., Hurley, J. F., Isaacman-VanWertz, G., Koss, A. R., Li, M., McKeen, S. A., Moshary, F., Peischl, J., Pospisilova, V., Ren, X., Wilson, A., Wu, Y., Trainer, M., and Warneke, C.: Volatile chemical product emissions enhance ozone and modulate urban chemistry, Proc Natl Acad Sci U S A, 118, 10.1073/pnas.2026653118, 2021.

Liu, T., Hong, Y., Li, M., Xu, L., Chen, J., Bian, Y., Yang, C., Dan, Y., Zhang, Y., and Xue, L.: Atmospheric oxidation capacity and ozone pollution mechanism in a coastal city of southeastern China: analysis of a typical photochemical episode by an observation-based model, Atmospheric Chemistry and Physics, 22, 2173-2190, 10.5194/acp-22-2173-2022, 2022.

Qin, M., Murphy, B. N., Isaacs, K. K., McDonald, B. C., Lu, Q., McKeen, S. A., Koval, L., Robinson, A. L., Efstathiou, C., and Allen, C.: Criteria pollutant impacts of volatile chemical products informed by near-field modelling, Nature sustainability, 4, 129-137, 10.1038/s41893-020-00614-1, 2021.

2) In this study only 38 VOCs species were detected, which was much less than the photochemical species requirements of PAMS and also not conducive to the operation of the OBM model. It is also necessary for the author to explain the quality control of the online VOCs monitoring instrument;

**Authors' reply:**

■ Thank you for your comment. The up-to-dated 2017 PAMS target list includes 27 priority compounds and 37 optional compounds (Addtional Revisions to the PAMS Compound Target List, last accessed 2024-Jan-30). We agree with you that the measured 38 VOCs species at Birmingham Supersite was much less than the PAMS requirements. Nevertheless, the measured species in this study covered 20 of the 27 priority compounds. Mixing ratios of alkanes, alkenes, aromatics, and oxygenated VOCs species with high OH reactivities were well-recorded at the sampling site. In terms of the model performance, the mixing ratios of modelled $O_3$ during the initial period, $O_3$ period, and clear-out period were significantly correlated with those of the observed $O_3$ ($R^2 = 0.9$ at all periods, $P < 0.05$).

From the perspective of $O_3$ formation mechanism, *in-situ* production of $O_3$ is driven by radical

chemistry. Therefore, even a simplified $NO_x$-$O_3$-VOC mechanism with limited numbers of VOCs input can describe the most important features of chemical formation of $O_3$ (Seinfeld and Pandis, 2016). For example, a recent study applied a box model constrained by $NO_x$ and a single compound, propane, to evaluate the chemical regime of urban $O_3$ in major cities in China, Japan, and the United States (Wolf et al., 2022). It was concluded that the elevated $O_3$ in Chinese cities and the slowed reduction of $O_3$ in Japan and US were likely attributed to decreased $NO_x$ emissions.

■ Regarding your comments on the quality control of the online VOCs monitoring instrument, This was completed regularly since the instrument is used for long-term monitoring and reports to national air quality networks. Direct calibration using 4 ppbv gas standard cylinders and the use of carbon response for this instrument was tightly controlled, with regular carbon-wise responses calculated for all species as a cross check. Calibration sequences of the 4 ppbv calibration used in this study are run at regular intervals, with responses analyzed as a function of time. FID response has been verified to be stable over the lifetime of the GC-FID instrument, and did not drift in any observable way over the sample analyzing period for this study. Detection limits for this instrument are not higher than 0.1 ppbv and are typically in the 5-10 pptv range. We have added Table 1 and Table 2 into the supplement material. Table 1 lists which species were directly calibrated, and which used equivalent carbon numbers for quantification. Table 2 lists effective carbon numbers of species which used carbon-wise responses. Additionally, Descriptions on GC-FID instrument has been fully revised, and the calibration sequence has been added in the revised manuscript onto **Page 6, Line 151-156**.

**Table 1.** Species quantified and their corresponding quantification method used in this study.

| Species | Quantification method |
| --- | --- |
| ethane | 4 ppbv gas standard cylinders |
| propane | 4 ppbv gas standard cylinders |
| i-butane | 4 ppbv gas standard cylinders |
| n-butane | 4 ppbv gas standard cylinders |
| i-pentane | 4 ppbv gas standard cylinders |
| n-pentane | 4 ppbv gas standard cylinders |
| 2-methylpentane | 4 ppbv gas standard cylinders |
| hexane | 4 ppbv gas standard cylinders |
| heptane | 4 ppbv gas standard cylinders |
| i-octane | 4 ppbv gas standard cylinders |
| ethene | 4 ppbv gas standard cylinders |
| propene | 4 ppbv gas standard cylinders |
| t-2-butene | 4 ppbv gas standard cylinders |
| 1-butene | 4 ppbv gas standard cylinders |
| c-2-butene | 4 ppbv gas standard cylinders |
| 1,3-butadiene | 4 ppbv gas standard cylinders |
| t-2-pentene | 4 ppbv gas standard cylinders |
| c-2-pentene | 4 ppbv gas standard cylinders |
| isoprene | 4 ppbv gas standard cylinders |
| acetylene | 4 ppbv gas standard cylinders |
| benzene | 4 ppbv gas standard cylinders |
| toluene | 4 ppbv gas standard cylinders |
| ethylbenzene | 4 ppbv gas standard cylinders |
| m-xylene | 4 ppbv gas standard cylinders |
| p-xylene | 4 ppbv gas standard cylinders |
| o-xylene | 4 ppbv gas standard cylinders |
| 1,3,5-trimethoxybenzene | 4 ppbv gas standard cylinders |
| 1,2,4-trimethoxybenzene | 4 ppbv gas standard cylinders |
| 1,2,3-trimethoxybenzene | 4 ppbv gas standard cylinders |
| cyclopentane | effective carbon number using toluene as reference |
| 3-methylpentane | effective carbon number using toluene as reference |
| nonane | effective carbon number using toluene as reference |
| i-butene | effective carbon number using toluene as reference |
| acetaldehyde | effective carbon number using toluene as reference |
| acetone | effective carbon number using toluene as reference |
| methanol | effective carbon number using toluene as reference |
| ethanol | effective carbon number using toluene as reference |

**Table 2.** Average concentration in ppbv and effective carbon number (ECN) of the measured VOCs at Birmingham Supersite over August, 2022.

|  | Species | Mean | SD | ECN |
|---|---|---|---|---|
| **Alkanes** | ethane | 1.69 | 1.48 | – |
|  | propane | 0.65 | 0.57 | – |
|  | i-butane | 0.27 | 0.27 | – |
|  | n-butane | 0.50 | 0.46 | – |
|  | cyclopentane | 0.03 | 0.04 | 5.00 |
|  | i-pentane | 0.19 | 0.17 | – |
|  | n-pentane | 0.09 | 0.10 | – |
|  | 2-methylpentane | 0.05 | 0.05 | – |
|  | 3-methylpentane | 0.03 | 0.03 | 5.00 |
|  | hexane | 0.03 | 0.03 | – |
|  | heptane | 0.02 | 0.02 | – |
|  | i-octane | 0.02 | 0.02 | – |
|  | nonane | 0.06 | 0.03 | 9.00 |
| **Alkenes** | ethene | 0.26 | 0.20 | – |
|  | propene | 0.10 | 0.08 | – |
|  | t-2-butene | 0.01 | 0.01 | – |
|  | 1-butene | 0.02 | 0.02 | – |
|  | i-butene | 0.02 | 0.01 | 4.00 |
|  | c-2-butene | 0.00 | 0.00 | – |
|  | 1,3-butadiene | 0.01 | 0.01 | – |
|  | t-2-pentene | 0.00 | 0.01 | – |
|  | c-2-pentene | 0.01 | 0.01 | – |
|  | isoprene | 0.12 | 0.13 | – |
| **Alkyne** | acetylene | 0.11 | 0.06 | – |
| **Aromatics** | benzene | 0.07 | 0.05 | – |
|  | toluene | 0.16 | 0.14 | – |
|  | ethylbenzene | 0.04 | 0.04 | – |
|  | m-xylene | 0.11 | 0.12 | – |
|  | p-xylene | 0.04 | 0.04 | – |
|  | o-xylene | 0.04 | 0.05 | – |
|  | 1,3,5-trimethoxybenzene | 0.01 | 0.01 | – |
|  | 1,2,4-trimethoxybenzene | 0.05 | 0.06 | – |
|  | 1,2,3-trimethoxybenzene | 0.01 | 0.02 | – |
| **OVOCs** | acetaldehyde | 1.09 | 0.59 | 1.00 |
|  | acetone | 2.21 | 1.13 | 2.00 |
|  | methanol | 3.72 | 2.35 | 0.75 |
|  | ethanol | 1.79 | 1.60 | 1.50 |

**References**

Seinfeld, J. H. and Pandis, S. N.: Atmospheric chemistry and physics: from air pollution to climate change, John Wiley & Sons, 2016.

Wolf, M. J., Esty, D. C., Kim, H., Bell, M. L., Brigham, S., Nortonsmith, Q., Zaharieva, S., Wendling, Z. A., de Sherbinin, A., and Emerson, J. W.: New insights for tracking global and local trends in exposure to air pollutants, Environmental Science & Technology, 56, 3984-3996, 10.1021/acs.est.1c08080, 2022.

3) Is the research based on case study representative in evaluating photochemical pollution in a certain region?

**Authors' reply:** Thank you for pointing this out. The Birmingham Supersite has been operated for many years and represents a typical UK urban background environment (Please see detailed description on surroundings of the site in revised manuscript **Page 4 Line 108-112**). As a case study in Birmingham, we evaluated the impacts of changes in VOC sources and speciation on the $O_3$ production. The evaluation is based on the national trend of VOC emissions that hold for Birmingham where on-road emissions, residential combustion, gas leakage, and non-industrial solvent usage are dominant sources for $O_3$ precursors. However, the results we obtained in this study may not hold for cities near large industrial VOC sources (i.e., oil and natural gas production (Edwards et al., 2014); steel and cement production (Yao et al., 2021)) can significantly affect composition and chemical reactivity of ambient VOCs. So whilst no location can every truly be considered as 'typical', this site is one that has been used for many years as being representative for a city with a mix of residential, business, energy and road traffic emission sources, and in turn is therefore very similar in nature to many other UK cities.

■ In responding to your comment, we have added "The attribution of VOCs sources based on the NAEI data can be thought of as representative for this case study as a typical urban environment, but it might not hold for cities near large industrial VOC sources (i.e., oil refinery and industrial production sites), since they can significantly affect composition and chemical reactivity of ambient VOCs." in the revised manuscript **Page 15 Line 379-383**.

**References**

Edwards, P. M., Brown, S. S., Roberts, J. M., Ahmadov, R., Banta, R. M., deGouw, J. A., Dube, W. P., Field, R. A., Flynn, J. H., Gilman, J. B., Graus, M., Helmig, D., Koss, A., Langford, A. O., Lefer, B. L., Lerner, B. M., Li, R., Li, S. M., McKeen, S. A., Murphy, S. M., Parrish, D. D., Senff, C. J., Soltis, J., Stutz, J., Sweeney, C., Thompson, C. R., Trainer, M. K., Tsai, C., Veres, P. R., Washenfelder, R. A., Warneke, C., Wild, R. J., Young, C. J., Yuan, B., and Zamora, R.: High winter ozone pollution from carbonyl photolysis in an oil and gas basin, Nature, 514, 351-354, 10.1038/nature13767, 2014.

Yao, S., Wang, Q., Zhang, J., Zhang, R., Gao, Y., Zhang, H., Li, J., and Zhou, Z.: Ambient volatile organic compounds in a heavy industrial city: Concentration, ozone formation potential, sources, and health risk assessment, Atmospheric Pollution Research, 12, 101053, 10.1016/j.apr.2021.101053, 2021

4) The research result showed that road transport played the most important VOC role in the case study ozone photochemistry despite it only contributing 3.3% of national VOCs emissions in 2019. Since the proportion of emissions from road transportation is so limited, how to further control?

**Authors' reply:** In support of Net Zero proposed by International Energy Agency, UK and many countries in the world delivered plans to promote electric-fuel hybrid and full-electric vehicles. UK government announced a commitment to phasing-out sales of new internal combustion engine vehicles by 2030. The target is set by 2035 in European countries. Government of China has been implementing policies on reduction/exemption of tax, as well as subsides on buying full-electric vehicles since 2012 (Wang et al., 2017). The widespread policy will over time lead to significant VOCs reductions from on-road transportation and related fuel (i.e., gasoline, natural gas, and liquefied petroleum gas (LPG)) usages.

We believe that this question is already well answered through Figures 5 and 6 and the Conclusion section (detailed discussion please see revised manuscript **Page 15-18**). In combination these demonstrate that further reducing road transport emissions, ultimately to 100% abatement, does continue to deliver further reductions in $P(O_3)$. Once 100% abated, however, the remaining

VOC sources such as solvents control P($O_3$) rates.

**References**

Wang, N., Pan, H., and Zheng, W.: Assessment of the incentives on electric vehicle promotion in China, Transportation Research Part A: Policy and Practice, 101, 177-189, 10.1016/j.tra.2017.04.037, 2017.

---

## Author Response (AR2)

Dear Dr. Eleanor:

Thank you for taking time to review our submission for the second-round. Yours and the reviewer's concerns on the modelling assumptions have been carefully addressed in the manuscript. This includes: (1) in the Introduction, literature reviews of relevance to the use of emission inventories in ozone modelling and analyses on decadal trends of ozone have been added; (2) in the Results and Discussion, descriptions of modelling assumptions have been elaborated, and limitations of the modelling scenario have been added. Revised text within the manuscript have been marked by red. Please see below as our point-by-point responses to the reviewer's comments.

Best Regards,

Jianghao Li (on behalf of all co-authors)

**Reviewer #1**

The manuscript titled "The impact of multi-decadal changes in VOCs speciation on urban ozone chemistry: A case study in Birmingham United Kingdom" by Jianghao Li and colleagues focuses on the effects of changes in the speciation of volatile organic compounds (VOCs) on urban ozone production. The study, utilizing a photochemical box model and data from the UK National Atmospheric Emission Inventory (NAEI), evaluates the shift in VOC sources and their photochemical reactivities from 1990 to 2019. Key findings include a significant reduction in VOC emissions from road transport and increased emissions from alcohols due to solvent use and industry processes.

Evaluating the impacts of emission reduction on tropospheric ozone from scientific and policy perspectives is essential. I believe a few more clarifications are necessary to assess this study's main conclusion adequately.

1)  The national emission inventory is utilized to interpolate the relative importance of different VOC sources and species in the historical context. Then, the analysis is applied to the *in -situ* observational dataset at a research site in Birmingham. I guess the study assumes that the emission sources and intensities directly correspond to each other between the local and national VOC emissions. It requires a thorough justification.

**Authors' reply:** In this study, we show the trends in anthropogenic VOC emissions from 1990 to 2019 estimated by the National Atmospheric Emission Inventory (NAEI). The emission reduction in VOCs from the anthropogenic sources described by NAEI were then developed into different modelling scenarios. We use this method to obtain the effect of emission reductions on ozone production sensitivity. According to the inventory, all observed pollutants investigated in this analysis can be almost entirely described by 6 source sectors (Table 1).

We agree that there is an overreaching assumption that the VOCs at the Birmingham Supersite are affected directly in the same proportion that VOCs are reported in national amounts in NAEI, and we reference this explicitly in the text. The Birmingham Supersite has been a major air quality research station operated for many years and is considered to represent a typical UK urban background environment. The analysis assumes that such a background location will reasonably

represent the national trends and distribution of VOC emissions for sources including on-road emissions, residential combustion, gas leakage, and non-industrial solvent usage. It should be noted however that the modelling completed here does not aim to exactly replicate the photochemistry of Birmingham, but to provide a starting point from which different sectors may have emissions reduced.

The technique that integrates emission inventories into box models has been commonly used in the research field of urban ozone pollution, since it provides a reasonable starting point for understanding how each inventory source sector may influence ozone production during a case study event. For example, Coggon et al. (2021) employed VOC emission flux from an emission inventory (fuel-based inventory of vehicle emissions with volatile chemical products) to evaluated contributions of each VOC emission sector to ozone production at an urban background site in New York City. Nelson et al. (2021) used emission inventories from Emission Database for Global Atmospheric Research (EDGAR) to investigate *in -situ* ozone production sensitivity to inventory source sectors at an urban site in Delhi.

As for the reviewer's concerns on using emission inventories in box models, we have specified limitations of this analysis in Line 410 – 414: The attribution of VOCs sources based on the NAEI data can be thought of as representative for this case study as a typical urban environment, but it might not hold for cities near large industrial VOC sources (i.e., oil refinery and industrial production sites), since they can significantly affect composition and chemical reactivity of ambient VOCs. In responding to your comment and to introduce the use of emission inventories in modelling studies, we have added above and additional relevant literature into the revised manuscript (For detailed information, please see Line 110 – 122).

References

Coggon, M. M., Gkatzelis, G. I., McDonald, B. C., Gilman, J. B., Schwantes, R. H., Abuhassan, N., Aikin, K. C., Arend, M. F., Berkoff, T. A., Brown, S. S., Campos, T. L., Dickerson, R. R., Gronoff, G., Hurley, J. F., Isaacman-VanWertz, G., Koss, A. R., Li, M., McKeen, S. A., Moshary, F., Peischl, J., Pospisilova, V., Ren, X., Wilson, A., Wu, Y., Trainer, M., and Warneke, C.: Volatile chemical product emissions enhance ozone and modulate urban chemistry, Proc Natl Acad Sci U S A, 118, 10.1073/pnas.2026653118, 2021.

Nelson, B. S., Stewart, G. J., Drysdale, W. S., Newland, M. J., Vaughan, A. R., Dunmore, R. E.,

Edwards, P. M., Lewis, A. C., Hamilton, J. F., and Acton, W. J.: In situ ozone production is highly sensitive to volatile organic compounds in Delhi, India, Atmospheric Chemistry and Physics, 21, 13609-13630, 10.5194/acp-21-13609-2021, 2021.

**Table 1.** Relative contributions (%) of ozone precursors emitted from the six emission inventory sectors.

| | road transport | fuel fugitive | agriculture | industrial process | combustion | solvents | SUM |
|---|---|---|---|---|---|---|---|
| **ethane** | 6.0 | 48.6 | 39.1 | 2.1 | 2.9 | 0.0 | 98.7 |
| **butanes** | 34.9 | 35.2 | 0.0 | 1.4 | 1.2 | 27.0 | 99.8 |
| **propanes** | 82.0 | 9.9 | 0.0 | 0.7 | 0.5 | 6.7 | 99.8 |
| **C>=6 alkanes** | 39.5 | 31.3 | 0.0 | 2.2 | 1.5 | 22.8 | 97.2 |
| **acetylene** | 85.8 | 7.6 | – | 2.7 | 0.0 | – | 96.2 |
| **ethene** | 8.6 | 86.9 | – | 4.5 | – | – | 100.0 |
| **butenes** | 96.1 | 0.7 | – | 0.7 | 1.5 | – | 99.0 |
| **propene** | 64.1 | 34.1 | – | 1.8 | – | – | 100.0 |
| **pentenes** | 100.0 | – | – | – | – | – | 100.0 |
| **1,3-butadiene** | 76.0 | 3.3 | – | 3.5 | 11.0 | – | 93.8 |
| **toluene** | 80.0 | 3.8 | 0.3 | 0.6 | 1.1 | 10.3 | 96.1 |
| **xylenes** | 72.0 | 1.3 | 0.3 | 1.0 | 1.3 | 21.6 | 97.6 |
| **other aromatics** | 71.3 | 2.9 | – | 1.8 | 5.3 | 12.6 | 94.0 |
| **acetaldehyde** | 69.0 | – | 0.2 | 13.0 | 0.0 | – | 82.1 |
| **acetone** | 17.0 | – | – | 15.4 | 0.2 | 65.6 | 98.3 |
| **methanol** | – | 0.0 | – | 3.0 | – | 96.8 | 99.8 |
| **ethanol** | 7.3 | 0.1 | 11.9 | 48.8 | 5.8 | 25.3 | 99.1 |
| **$NO_x$** | 33.3 | – | 3.9 | 18.4 | 28.0 | – | 83.5 |
| **CO** | 14.5 | 1.2 | – | 32.3 | 34.0 | – | 82.0 |

2) Emission changes do not necessarily occur directly in ambient concentration distributions as the atmospheric lifetime of different VOC species widely varies. The direct scaling of ambient VOCs by reflecting emission changes may be flawed.

**Authors' reply:** Ambient distribution in VOC speciation and concentration are affected by photochemical loss, anthropogenic/natural emissions, and advection process. It is inevitably a challenge to produce a concise representation of sectors and emissions when the underlying complexity of influence factor is very high. However from the perspective of ozone, the effect of reductions in $O_3$ precursors by sources can be obtained by varying source sectors from emission inventories. We developed model scenarios that vary model constrained concentrations according to their contributions to sources in the NAEI emission inventory. Although this may not accurately reflect how at fine geographic scales individual VOCs are distributed, it does adequately capture the ozone change since this is formed over multi-hour time scales and hence regional domains. The modelling is focused on reproducing integrated ozone production changes, not localized radical concentrations. It the paper was attempting to simulate the latter, then the reviewer's concern would be more relevant.

In the revised manuscript (Line 422 – 424), we have elaborated on the term 'emission change': …For this analysis, emission changes in these sectors can be obtained by reducing model constrained concentrations of VOCs, $NO_x$, and CO according to their contributions arising from individual emission sectors. At the end of the analysis, we have added limitations of the analysis (Line 439 – 445): It is important to acknowledge the limitation of this analysis. In the real world, reductions in ambient VOCs from the NAEI sectors are affected by photochemical loss rate and advection processes, potentially altering the proportion of VOCs that would be observed with each sector reduction at the measurement site. This would potentially be an important consideration if instantaneously radical budgets were being evaluated, but it is a less significant issue when integrated ozone production effects.

3) The underlying assumption is that all other natural conditions are identical and that only anthropogenic emissions have changed over time. This needs to be justified as natural $NO_x$ emissions and mainly BVOC emissions might have changed. In addition, a broad discussion of how weather conditions have changed in the ozone photochemical context.

**Authors' reply:** Thank you for pointing this out. Ozone mixing ratios are largely influenced by precursor emission, surface regional transport, deposition, and stratosphere-atmosphere exchange. The interactions among ozone precursors play an important role in urban atmosphere. To implement successful ozone reduction strategies, a good understanding of the non-linear processing of its precursor is imperative. Hence, we focus on the impacts of VOC emissions in the context of historical changes on urban ozone production in this study.

In responding to your comments on influences of biogenic emissions and weather patterns, we have added literature reviews in the Introduction (revised manuscript Line 93 – 109) to discuss previous analyses on decadal trends of ozone. Here we briefly describe research that focuses on that published literature: Previous model analysis of decadal trends of ozone has centred on the association between extreme weather and ozone events and projected changes in ozone concentration under chemical regime change scenarios. This includes (1) understanding the role of anticyclonic conditions and higher temperature during late spring and summer in biogenic emissions and elevated ozone concentrations (Diaz et al., 2020; Hertig et al., 2020; Lewis et al., 2021; Finch and Palmer, 2020); (2) obtaining ozone sensitivities to VOCs, $NO_x$, and aerosol (Ivatt et al., 2022; Gouldsbrough et al., 2024); (3) investigating potential impacts of future climate change on occurrence of ozone event (Gouldsbrough et al., 2022; Liu et al., 2022).

References

Diaz, F. M., Khan, M. A. H., Shallcross, B. M., Shallcross, E. D., Vogt, U., and Shallcross, D. E.: Ozone trends in the United Kingdom over the last 30 years, Atmosphere, 11, 534, 10.3390/atmos11050534, 2020.

Finch, D. P. and Palmer, P. I.: Increasing ambient surface ozone levels over the UK accompanied by fewer extreme events, Atmospheric environment, 237, 117627, 10.1016/j.atmosenv.2020.117627, 2020.

Gouldsbrough, L., Hossaini, R., Eastoe, E., and Young, P. J.: A temperature dependent extreme value

analysis of UK surface ozone, 1980–2019, Atmospheric Environment, 273, 118975, 10.1016/j.atmosenv.2022.118975, 2022.

Gouldsbrough, L., Hossaini, R., Eastoe, E., Young, P. J., and Vieno, M.: A machine learning approach to downscale EMEP4UK: analysis of UK ozone variability and trends, Atmospheric Chemistry and Physics, 24, 3163-3196, 10.5194/acp-24-3163-2024, 2024.

Hertig, E., Russo, A., and Trigo, R. M.: Heat and ozone pollution waves in central and south Europe—characteristics, weather types, and association with mortality, Atmosphere, 11, 1271, 10.3390/atmos11121271, 2020.

Ivatt, P. D., Evans, M. J., and Lewis, A. C.: Suppression of surface ozone by an aerosol-inhibited photochemical ozone regime, Nature Geoscience, 15, 536-540, 10.1038/s41561-022-00972-9, 2022.

Lewis, A. C., Allan, J. D., Carruthers, D., Carslaw, D. C., Fuller, G. W., Harrison, R. M., Heal, M. R., Nemitz, E., Reeves, C., Williams, M., Fowler, D., Marner, B. B., Williams, A., Carslaw, N., Moller, S., Maggs, R., Murrells, T., Quincey, P., and Willis, P.: Ozone in the UK: Recent Trends and Future Projections. Air Quality Expert Group. https://uk-air.defra.gov.uk/library/reports.php?report_id=1064, 2021, last assess: 27 March 2024.

Liu, Z., Doherty, R. M., Wild, O., O'connor, F. M., and Turnock, S. T.: Tropospheric ozone changes and ozone sensitivity from the present day to the future under shared socio-economic pathways, Atmospheric Chemistry and Physics, 22, 1209-1227, 10.5194/acp-22-1209-2022, 2022.